# Effectiveness of Vision Transformer for Fast and Accurate Single-Stage Pedestrian Detection

**Jing Yuan**
Department of Electric and Electronic Engineering
Imperial College London
j.yuan20@imperial.ac.uk

**Panagiotis Barmpoutis**
Department of Computer Science
University College London
p.barmpoutis@ucl.ac.uk

**Tania Stathaki**
Department of Electric and Electronic Engineering
Imperial College London
t.stathaki@imperial.ac.uk

## Abstract

Vision transformers have demonstrated remarkable performance on a variety of computer vision tasks. In this paper, we illustrate the effectiveness of the deformable vision transformer for single-stage pedestrian detection and propose a spatial and multi-scale feature enhancement module, which aims to achieve the optimal balance between speed and accuracy. Performance improvement with vision transformers on various commonly used single-stage structures is demonstrated. The design of the proposed architecture is investigated in depth. Comprehensive comparisons with state-of-the-art single- and two-stage detectors on different pedestrian datasets are performed. The proposed detector achieves leading performance on Caltech and Citypersons datasets among single- and two-stage methods using fewer parameters than the baseline. The log-average miss rates for Reasonable and Heavy are decreased to 2.6% and 28.0% on the Caltech test set, and 10.9% and 38.6% on the Citypersons validation set, respectively. The proposed method outperforms SOTA two-stage detectors in the Heavy subset on the Citypersons validation set with considerably faster inference speed.

## 1 Introduction

Pedestrian detection is a popular task subordinate to object detection in computer vision. This task aims to locate and classify pedestrians in images or videos accurately. Pedestrian detection is very important as it serves as the prerequisite of various vision tasks [1], such as human-centric tasks (person re-identification [2, 3], person search [4], human pose estimation [5] etc.) and more generic multi-object tracking [6]. It has been applied to autonomous driving [7, 8], video surveillance [9] and action tracking. In this paper, we focus on the detection based on RGB images.

Pedestrian detection suffers from significant occlusion and varying scales. Intra- and inter-class occlusion occur when a pedestrian is occluded by other pedestrians or objects like cars, bicycles etc. Both significantly reduce the discriminative features and destroy the regular shape of pedestrians. For varying scales, large pedestrians tend to have more informative features. Still, they are difficult to fully extract from a vast region, while features of small targets are compact but relatively ambiguous with less preserved details. In summary, the fluctuating amount and varying shape of effective features are the core problems. They challenge the capabilities of feature extraction modules, which act as a long-standing bottleneck in pedestrian detection.

36th Conference on Neural Information Processing Systems (NeurIPS 2022).

To deal with these problems, two-stage methods [10–15] based on Fast [16] and Faster R-CNN [16] have pervaded pedestrian detection tasks owing to the high detection accuracy. These methods first make coarse predictions of targets via the Region Proposal Network (RPN), then refine the bounding boxes and predict the final scores based on the features inside the proposal regions. In addition to methods for general object detection, pedestrian detectors take advantage of unique characteristics of targets, such as the mask of visible parts [17, 10] and key points of human bodies [18]. However, the inference speed of these methods is limited by the repeated predictions which makes them hard to be applied to real-world scenes.

To achieve faster inference, single-stage approaches, which only make one round prediction, are developed [19] and applied to pedestrian detection [20–24]. However, they suffer from decreased detection accuracy. The miss rates of two-stage methods in the Reasonable subset on Caltech are reduced to less than 4% [14, 10, 25, 18], while those for single-stage methods are larger than 4.5% [22]. For the Citypersons validation set, the miss rates for the former are less than 40% in the Heavy subset, which is much lower than the latter (42% [24]). In this paper, we aim to improve the detection accuracy, especially in Reasonable and Heavy subsets, and to narrow the gap between single- and two-stage methods with fast inference.

Typical single-stage detectors use anchors (SSD [26]) or are anchor-free (CSP [22]). The former generates rectangular bounding boxes with different aspect ratios and scales centered at each pixel of the feature maps at certain levels. Anchor scales are designed to be smaller at lower levels to facilitate the detection of small objects. These methods predict the offsets w.r.t. the upper left position, height and width of the anchor. Taking CSP as an example, the latter method only predicts the logarithm height and offsets w.r.t. centers of each pixel. For anchor-based single-stage detectors, ALFNet [21] refines anchors progressively with stacked prediction blocks to remedy the lack of proposal regions.

In the past year, most research has focused on the fusion of representative features [27–32] to improve single-stage methods. For example, [30] enhances features via increasing semantic information at a low level and enriches the localization information at a high level. Similarly, [32] fuses the feature maps with different scales in adequate proportions. [29, 31] also explore new strategies to aggregate multi-level features. These feature enhancements are mainly performed along the dimension of feature level due to the intrinsically unbalanced feature information between shallow and deep feature levels. However, this unbalance also exists in two-stage detectors. As such, this is not the particular reason for the poorer accuracy of single-stage methods.

The general architecture of single- and two-stage detectors are compared in Figure 1. Assuming that the training strategies and the detection head are the same for both methods, the difference in the architecture lies in the information fed into the detection head. For two-stage methods, both positions and features of the proposal regions are fed into the detection head. These proposals contain potential pedestrians. Thereby, the detection head classifies spatially target-focused features with fewer background interruptions and refines the bounding boxes by predicting small offsets from the proposal positions. For single-stage methods, each pixel in the feature map serves as the 'proposal region' with no pre-estimated positions. The receptive fields of these pixels share the same size, which may be too small to include sufficient information for large targets or so large that the background information overwhelms the useful features. This is more challenging for the classifier compared to two-stage methods. Additionally, single-stage methods have to regress from scratch, which is more difficult than simple refinement. Thus, the lack of spatially target-focused feature representation and the prediction of bounding boxes from scratch are the two key bottlenecks hindering the improvement of single-stage detectors.

To make the features fed into the detection head concentrate on the targets or other helpful information automatically without the assistance of proposal regions, we take advantage of vision transformers in this paper. Vision transformers describe the pairwise dependency of each entity in the feature map with attention weights. The output weighted averaged feature is the adaptive aggregation of important entities (with higher attention weights) while the disturbing information (with lower attention weights) is suppressed. Using such attention mechanism on top of the backbone enables the single-stage detector to supplement spatially filtered features easily for subsequent classification and regression. In this case, the modified detector makes the best use of the fast inference originating from single-stage methods and more effective features. Our main contributions are as follows:

- Demonstrate the effectiveness of the deformable vision transformer in improving the accuracy of commonly used single-stage detectors on pedestrian datasets.

- Extend the application of vision transformers on top of the backbone in pedestrian detection tasks.
- Achieve the best performance among single-stage detectors on the Caltech test set and Citypersons validation set while maintaining fast inference and reducing the number of parameters.
- Narrow the gap of detection accuracy between the single- and two-stage methods in pedestrian detection.

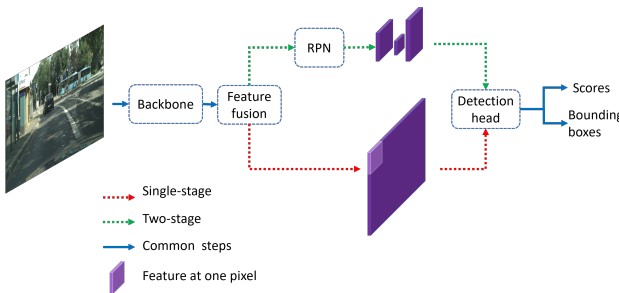

Figure 1: Generic detection pipeline of the single- and two-stage detectors. The detection head consists of two parallel branches for classification and regression.

## 2   Related works

Currently, vision transformers are used to establish general-purpose backbones (ViT [33] and Swin Transformer [34]) or stack on top of the backbone (DETR [35]). Since we focus on pedestrian detection, this paper explores the latter case. DETR consists of the convolution backbone, six encoders and decoders and the prediction head. It is an inspiring end-to-end detector but is memory-consuming. DETR requires massive memory to store the self-attention weights within each Multi-Head Self-Attention (MHSA) layer. The memory cost is linear to the number of attention heads and is square to the number of pixels in the down-sampled feature map. Additionally, first and second-order momentums in optimization introduce further memory cost in the training procedure. In pedestrian detection, more attention heads and relatively large down-sampled feature maps are preferred to enhance detection accuracy, especially for small targets. This results in memory explosion using DETR. To this end, deformable DETR [36] is proposed. It only needs the attention weights at several sampling locations rather than each pixel in the feature map. The memory cost of the attention weights is linear to the number of pixels, which makes training with high resolutions possible. Experiments show that the deformable DETR outperforms the Faster R-CNN [37] and DETR on COCO 2017 validation set [38].

So far, vision transformers show great potential, but they have rarely been applied to pedestrian detection in the form of DETR or its variants. This is because it has been observed that they perform worse than the commonly used Faster R-CNN on CrowdHuman dataset [39] and require tenfold training time [40]. Although [40] proposed using dense object queries and the rectified attention field to enhance scale-adaptive feature extraction in the decoding phase, the modified deformable DETR still shows a limited advantage over the traditional Faster R-CNN. This implies that rigidly putting the whole six encoder-decoder pairs into the pedestrian detector may not be cost-effective. As an example, BoTNet [41] only substitutes the convolution layers in residual bottleneck blocks in the last stage of ResNet [42] for MHSA layers, but it produces strong performance on ImageNet validation set [43]. Inspired by this and the limitation above, our work only utilizes a single encoder of the deformable vision transformer as an adaptive feature extractor and applies it to commonly used single-stage detectors for better detection accuracy and fast inference.

## 3   Method

**Deformable Vision Transformer Encoder**: The deformable vision transformer encoder (Figure 2) takes the $L$ feature maps $\left\{z^l\right\}_{l=1}^{L}$ with height $H_l$ and width $W_l$ at scale $l$ and reference points,

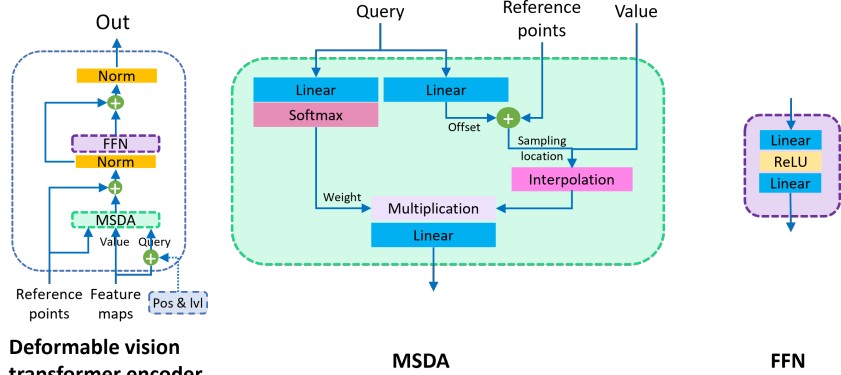

Figure 2: **Left**: The structure of the deformable vision transformer encoder. It is the same as the encoder in DETR except that the key tensor is removed while reference points are required. **Mid**: The MSDA averages the features at sparse sampling locations across different scales with weights computed from the query tensor. **Right**: Feed-Forward Network (FFN).

which are the positions of grid centers of the feature maps, as inputs. It outputs the enhanced feature maps with the same resolution as the input. The input feature maps are first added by fixed encoded positional [35] and learnable level information [36] to disambiguate spatial and scale positions, then projected to the query feature map $z_q$ via a linear layer. The feature maps also generate the value feature maps $z_v$ with a linear layer but without encoding. The query feature map $z_q$, value feature map $z_v$ together with the pre-generated reference points are sent to the Multi-Scale Deformable Attention (MSDA) layer to enhance spatially adaptive features, followed by a Feed-Forward Network (FFN). In summary, the encoder supplements the semantic information of input feature maps via the embedded attention layer and FFN.

**Multi-Scale Deformable Attention**: The MSDA layer sums the selected entities at sampling locations in the value feature map $z_v$ with predicted attention weights by each corresponding query entity. These attention weights $W$ are the linear projection of query features $z_q$ followed by a softmax operator along scale and sampling point dimensions. For a single query entity, it only needs $N_h N_l N_p$ attention weights, representing the significance of selected value features at different attention heads, scales, and points. Selections are decided by the sampling locations which are the summation of reference points $p$ and sampling offsets $\Delta p$ which is the embedding of the query features. At each float sampling location, the selected value feature is bilinear interpolated for accuracy and training offset predictor. With weight, sampling locations and selected value features prepared, the $q$-th element of the separate output feature $z^{so,h} \in \mathbb{R}^{N_q \times c_v}$ ($N_q = \sum_{l=1}^{L} H_l W_l$, $c_v$ is the number of channels ) at attention head $h$ (total $N_h$ heads) is

$$z_q^{so,h} = \sum_{p=1}^{N_p} \sum_{l=1}^{L} W_{plhq} v_{p_{ql} + \Delta p_{plhq}} \tag{1}$$

where $p$, $q$, $l$ and $h$ index the sampling offsets, elements of the deformable attention feature $z^o$, the scale of value $v$ and attention head. $W_{plhq}$ is a value from the weight $W \in \mathbb{R}^{N_q \times N_h \times L \times N_p}$. $p \in \mathbb{R}^{N_q \times L \times 2}$ and $\Delta p \in \mathbb{R}^{N_q \times N_h \times L \times N_p \times 2}$ are the reference points and sampling offsets. $p_{ql}$ and $\Delta p_{plhq}$ denote the position of a single reference point and one of its corresponding sampling offsets respectively. The separate output features from $N_h$ attention heads are projected to the $q$-th element of the final output deformable attention feature $z^o$ by a linear layer expressed as

$$z_q^o = \sum_{h=1}^{N_h} W_h' z_q^{so,h} \tag{2}$$

where $W_h' \in \mathbb{R}^{c \times c_v}$ denotes the learnable weight for the $h$-th attention head.

**Proposed Feature Enhancement Module**: The features are enhanced owing to the self-attention mechanism to supplement the spatially adaptive features across multiple scales. The proposed module

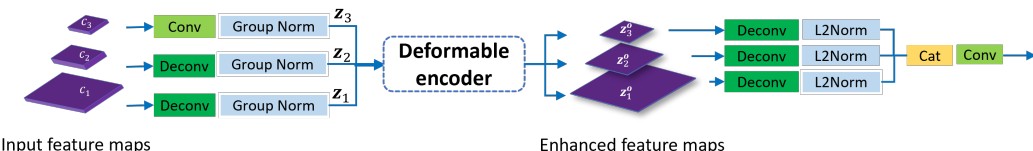

Figure 3: The overall architecture of the proposed feature enhancement module.

simply consists of convolution/deconvolution and normalization layer pairs ahead of and after the deformable encoder and a final feature fusion step as illustrated in Figure 3. In this module, input feature maps from the backbone are first upsampled with deconvolution layers or encoded with the convolution layer to generate multi-scale feature maps $\left\{z^l\right\}_{l=1}^3$. They are followed by group normalization to prevent the Internal Covariate Shift (ICS) that might be inducted by subsequent linear operations in the deformable encoder. The encoder yields enhanced multi-scale feature maps. Enhanced features are upsampled to keep the resolution as $(H/4, W/4)$ for accurate detection. They are normalized via L2Norm [22] before concatenation along the channel dimension. This makes the features at different scales contribute equally to the final feature representation fed into the detection head. Concatenated features are compressed along the channel dimension to reduce the network parameters. The output feature maps can be fed into the detection head used in SSD, CSP etc. Except for this standard structure, the use of convolution/deconvolution layers can be adjusted according to the resolution of input, and the concatenation step can be removed if predictions are made at separate levels.

**Training**: For anchor-free cases, namely CSP, the loss function follows [22]. The overall loss consists of three parts as Equation(3) where $L_c$, $L_h$ and $L_o$ stand for the center heatmap loss, height map loss and offset loss. Weights for each loss $\lambda_c$, $\lambda_s$ and $\lambda_o$ are set as 0.01, 1 and 0.1 [22]. For anchor-based cases, namely SSD or ALFNet, the multi-task loss function is formulated with two objectives as Equation(4) [21] where $\lambda_{cls}$ is experimentally set as 0.01 in the following experiments.

$$L_{af} = \lambda_c L_c + \lambda_s L_h + \lambda_o L_o \tag{3}$$

$$L_{ab} = \lambda_{cls} L_{cls} + L_{loc} \tag{4}$$

**Inference**: For anchor-free single-stage methods, the predicted width is the height multiplied by the uniform aspect ratio 0.41 [22]. If not specified, bounding boxes with scores above 0.01 are kept and merged by Non-Maximum Suppression (NMS) with the IoU threshold of 0.5.

## 4 Experiments

### 4.1 Settings

**Datasets**: The proposed detector is evaluated on two commonly used public pedestrian datasets: **Caltech** [44] and **Citypersons** [45]. The Caltech dataset is an approximately 10 hours of 480x640 video taken in a single urban city. The standard training set contains about 250k frames with 350k bounding boxes. In our experiments, the training data augmented by 10 folds containing 42782 images with 13674 persons and the standard test set containing 4024 images with corresponding new annotations [46] and fixed aspect-ratio for bounding boxes [47] are used. Citypersons training set recorded across 18 different cities, 3 seasons and various weather conditions with 19654 persons in 2975 high resolution (1024x2048) images. The validation set contains 500 images across 3 cities.

**Training Details**: If not specified, the ResNet50/VGG16 pre-trained on ImageNet, Adam with moving average weights [48] and step learning rate schedule are applied. Data augmentation techniques including random horizontal flips with a probability of 0.5 and scaling are applied. For Caltech, additional random color distortion and cropping are implemented. The input images are rescaled to 336x448 and 640x1280 for Caltech and Citypersons datasets respectively. The detectors are trained with a single NVIDIA GeForce RTX 3090 GPU for 10 and 75 epochs with batch size 16 and 4 on Caltech and Citypersons respectively using the anchor-free CSP detection head. The base learning rate is 0.5e-4 and decreased by a factor of 0.5 after 6 and 60 epochs respectively. Initialization is performed with a randomly chosen and fixed seed. Tools provided by [47] are used in the experiments.

**Metrics**: Log-average Miss Rate (denoted as $MR^{-2}$ or miss rate in this paper) over False Positive Per Image (FPPI) in the range $[10^{-2}, 10^0]$ is calculated over Reasonable, Small, Heavy, and All subsets defined in Table 1. The lower the $MR^{-2}$ the better.

Table 1: The experimental settings of four subsets.

| Subset | Reasonable | Small | Heavy | All |
|---|---|---|---|---|
| Height | [50, inf] | [50, 75] | [50, inf] | [20, inf] |
| Visibility | [0.65, inf] | [0.65, inf] | [0.2, 0.65] | [0.2, inf] |

## 4.2 Ablation Experiments

**Effectiveness of Enhanced Feature Maps**: For generality, the proposed module is applied to three commonly used single-stage structures in pedestrian detection: anchor-based, progressive refinement, and anchor-free.

For anchor-based single-stage methods, like SSD300 [26], Table 2 shows that the proposed module shows stable and significant improvement in all three subsets with an IoU threshold of 0.5 in NMS.

For refined anchors, we append two Convolutional Predictor Blocks [21] after the feature maps at the last three stages of the backbone and an extra stage. The second block refines the coarse anchors predicted by the first block. Table 3 demonstrates that even though the anchors are refined progressively to remedy the lack of proposal regions, the separate enhancement at each level can bring improvement in certain subsets (level 0 improves Reasonable and All subsets, level 1, 2 improves Heavy subset). The combination of multi-level inputs brings in stable improvement in all the subsets.

For anchor-free methods, We evaluated the influence of the proposed module on the baseline CSP [22] detector on Caltech (Table 4) and Citypersons (Table 5). Results from both datasets indicate that with the enhanced feature maps, the miss rates are decreased significantly in Reasonable, Heavy and All subsets, in particular, the Heavy subset witnesses a decrease of up to 7.9%, and the Reasonable subset up to 3.1%. The additional enhancement module increases the inference time, however, they contain fewer learnable parameters as shown in Table 6. As the combination of CSP achieves the best results, subsequent experiments follow this implementation.

According to the above, the proposed module is effective for general single-stage detectors on pedestrian datasets, owing to the multi-scale deformable self-attention mechanism to enhance spatially adaptive features across levels.

Table 2: Comparison of the SSD300 detector w/o the proposed feature enhancement module on Caltech test set with input size 300x300.

| Methods | IoU | Enhanced feature levels | | | | | Reasonable | Heavy | All |
|---|---|---|---|---|---|---|---|---|---|
| | | 1 | 2 | 3 | 4 | 5 | | | |
| SSD300 | 0.5 | | | \ | | | 28.6 | 70.7 | 72.3 |
| | 0.25 | | | | | | 27.1 | 71.3 | 71.2 |
| + Enhancement module | 0.5 | | | | | ✓ | 24.9 | 69.3 | 71.0 |
| | 0.25 | | | | | ✓ | 26.2 | 70.2 | 71.3 |
| | 0.5 | | | | ✓ | | 25.1 | 68.4 | 71.3 |
| | 0.25 | | | | ✓ | | 26.9 | 69.1 | 71.6 |
| | 0.5 | ✓ | ✓ | ✓ | | | 22.6 | 68.5 | 71.0 |
| | 0.25 | ✓ | ✓ | ✓ | | | 24.9 | 67.7 | 71.2 |
| | 0.5 | ✓ | ✓ | | | | 23.9 | 68.4 | 72.2 |
| | 0.25 | ✓ | ✓ | | | | 25.6 | 69.1 | 72.6 |
| | 0.5 | ✓ | | | | | 23.5 | 68.7 | 71.4 |
| | 0.25 | ✓ | | | | | 24.4 | 71.0 | 71.8 |

**Feature Maps Scales**: Different combinations of multi-scale feature maps $\{z^l\}_{l=1}^3$ with three downsampling ratios (1/4, 1/8 and 1/16) are compared in Table 7. In this comparison, only $z_3^o$

Table 3: Comparison of the progressive refinement detector w/o the proposed feature enhancement module on Caltech test set with input size 480x640.

| | Enhanced feature levels | | | Reasonable | Heavy | All |
|---|---|---|---|---|---|---|
| | 1 | 2 | 3 | | | |
| Progressive refinement | | \ | | 13.4 | 61.3 | 61.9 |
| + Enhancement module | ✓ | ✓ | ✓ | 12.3 | 60.8 | 61.2 |
| | | | ✓ | 13.9 | 59.2 | 61.7 |
| | | ✓ | | 13.5 | 60.0 | 62.7 |
| | ✓ | | | 12.4 | 61.5 | 59.8 |

Table 4: Comparison of the baseline CSP detector and the proposed detectors with the enhancement module on Caltech test set. FPS stands for frames per second evaluated during test process with input size 480x640.

| | Reasonable | Heavy | All | FPS |
|---|---|---|---|---|
| CSP | 6.8 | 50.7 | 62.3 | 39.2 |
| + Enhancement module | 3.7 (3.1↓) | 42.8 (7.9↓) | 56.97 (5.3↓) | 29.5 |

which has the same resolution as $z^3$ is upsampled to $(H/4, W/4)$, and is fed into the detection head without intermediate concatenation and L2Norm. Results show that feature maps with ratios $1/4$, $1/8$, and $1/6$ for each scale produce the best performance, which is yielded by upsampling $c_1$ and $c_2$ by two times while maintaining the scale of $c_3$.

**Enhanced Feature Map Scales**: Fix the downsampling ratios of $\{z^l\}_{l=1}^3$ as $1/4$, $1/8$, and $1/16$, feed different collections of multi-scale enhanced feature maps $\{z_l^o\}_{l=1}^3$ to the detection head. Note that the deformable encoder keeps the resolutions of the input feature maps, the downsampling ratios for $\{z_l^o\}_{l=1}^3$ are $1/4$, $1/8$ and $1/6$ respectively. All the enhanced feature maps are first upsampled to $(H/4, W/4)$ if needed, followed by normalization when multiple feature maps are utilized. They are then processed in three ways: 1. Cat: Concatenate them along the channel dimension followed by a compression layer to reduce the number of channels to 256. 2. Add: Implement element-wise sum of the enhanced feature maps. 3. Sep: No fusion operations across multi-scales; send them separately to the detection head which doubles the number of predictions. Table 8 compares various strategies to fuse the multi-scale enhanced feature maps. It shows that concatenation followed by L2Norm produces the overall best results in both Reasonable and Heavy subsets on the Caltech test set.

**Choice of the Normalization Method Applied to** $\{c^l\}_{l=1}^3$: As Table 8 shows, GN performs best in the Reasonable subset while L2N in the Heavy subset. Considering that the miss rate of GN is 11.6% lower in the Reasonable subset and only 1.2% higher in the Heavy subset compared to L2N, GN is utilized in our experiments before the encoder as presented in Figure 3.

**Number of Encoders**: Based on the settings in the last part, different numbers of encoders are tested. These encoders are connected in series. Table 9 presents that although more encoders can provide higher-level semantic information, the best result is observed when only a single encoder is applied. This phenomenon supports the design of BotNet [41] to some extent, which indicates that following the whole set of six encoders and decoders in (deformable) DETR may not be suitable for specific

Table 5: Comparison of the baseline CSP detector and the proposed detectors with the enhancement module on Citypersons validation set. FPS stands for frames per second. FPS stands for frames per second evaluated during test process with input size 1024x2048.

| | Reasonable | Small | Heavy | All | FPS |
|---|---|---|---|---|---|
| CSP | 11.7 | 14.4 | 41.8 | 38.2 | 8.2 |
| + Enhancement module | 10.9 (0.8↓) | 13.7 (0.7↓) | 38.6 (3.2↓) | 37.2 (1.0↓) | 6.8 |

Table 6: The number of learnable parameters in the baseline and proposed detectors.

|  | Backbone | Neck | Head | Total Param |
|---|---|---|---|---|
| CSP | 23.51 M | 14.68 M | 1.77 M | 39.96 M |
| + Enhancement module | 23.51 M | 8.73 M | 1.03 K | 32.24 M |

Table 7: Combinations of different resolutions of input feature maps fed into the encoder.

| $z^1$ | $z^2$ | $z^3$ | Reasonable | Heavy |
|---|---|---|---|---|
| 1/4 | 1/8 | 1/16 | **4.1** | **44.7** |
| 1/4 | 1/16 | 1/16 | 4.2 | 45.9 |
| 1/8 | 1/8 | 1/16 | 4.6 | 48.5 |
| 1/4 | 1/8 | 1/8 | 5.9 | 47.8 |

object detection tasks. A single encoder can also work effectively with the least number of learnable parameters, which prevents overfitting.

## 4.3 Comparison with the state-of-the-arts

Table 10 and Table 11 show that the proposed module achieves the lowest miss rates in Reasonable and Heavy subsets and leading performance in the All subset on both datasets among presented single-stage detectors.

For the Caltech dataset, the lowest miss rate (3.7%) of the proposed detector in the Reasonable subset is 0.2% smaller than that of the two-stage KGSNet presented in the upper part of Table 10. With pretraining on the Citypersons dataset, the miss rates on all three subsets of the Caltech dataset are reduced significantly and reach the lowest compared to other pretrained detectors as shown in the bottom part of Table 10. For the Citypersons dataset, the proposed two detectors even outperform the competitive two-stage detectors in the Reasonable and especially Heavy subset with a miss rate of 38.6% which is 1.1% lower than that of the best of two-stage detectors.

Overall, with the enhanced spatially adaptive and multi-scale features, the gap between single- and two-stage detectors in Reasonable and Heavy subsets on different detectors has been narrowed. Surprisingly, the proposed single-stage method outperforms the accurate two-stage methods on certain pedestrian datasets, such as the Citypersons dataset. It should be noted that two-stage methods usually produce overall better accuracy than single-stage methods with the advantage of the region proposal network and refined bounding boxes and at the expense of inference time. Apart from accuracy, fast inference also matters for pedestrian detection in practical scenes. The combination of CSP and the proposed module has a simple structure, which is easy to perform and effective. On the contrary, the leading two-stage KGSNet, for example, takes advantage of the additional proposal

Table 8: Combinations of enhanced feature maps with different scales in different ways followed by three Normalization Methods (NM). L2N, LN and GN stand for L2Norm, Layer Normalization and Group Normalization respectively. [1] Train boldly without any learning rate schedules. [2] Normalization methods applied to the enhanced feature maps after deconvolution. [3] Normalization methods applied to the feature maps produced by the backbone before being sent to the encoder.

| | | | | | | | | | | | | | | |
|---|---|---|---|---|---|---|---|---|---|---|---|---|---|---|
| $z_1^o$ | ✓ | | | ✓ | ✓ | ✓ | ✓ | ✓ | ✓ | ✓ | ✓ | ✓ | ✓ | ✓ |
| $z_2^o$ | | ✓ | | | | | | | | | | | | ✓ |
| $z_3^o$ | | | ✓ | ✓ | ✓ | ✓ | ✓ | ✓ | ✓ | ✓ | ✓ | ✓ | ✓ | ✓ |
| Cat | | | | | | | ✓ | ✓ | ✓ | ✓ | ✓ | ✓ | ✓ | ✓ |
| Add | | | | ✓ | ✓ | | | | | | | | | |
| Sep | | | | | | ✓ | | | | | | | | |
| NM[2] | | | | | L2N | L2N | | GN[1] | LN[1] | L2N[1] | L2N | L2N | L2N | L2N |
| NM[3] | GN | GN | GN | GN | GN | GN | GN | GN | GN | GN | GN | L2N | BN | GN |
| Reasonable | 4.3 | 5.7 | 4.1 | 7.0 | **3.7** | 4.2 | 5.9 | 6.4 | 4.0 | 3.9 | **3.7** | 4.3 | 4.5 | 5.3 |
| Heavy | 44.5 | 48.8 | 44.7 | 51.3 | 45.5 | 48.1 | 50.7 | 49 | 47.4 | 44.5 | 42.8 | **42.6** | 44.6 | 44.1 |

Table 9: Using 1 to 4 encoders in the detection neck.

| Number of encoders | 1 | 2 | 3 | 4 |
|---|---|---|---|---|
| Reasonable | **3.7** | 4.9 | 4.8 | 5.5 |
| Heavy | **42.8** | 46.6 | 43.1 | 45.1 |

Table 10: Comparison with the state-of-the-art single- and two-stage pedestrian detectors on Caltech test set. [†] Results reported by [22]. Bottom part: Pretrained on Citypersons and tested on Caltech. The best and second-best results are in bold and underlined with red for two-stage methods (upper part) and black for single-stage methods.

| Method | Backbone | Stage | Reasonable | Heavy | All |
|---|---|---|---|---|---|
| Faster R-CNN[†] [37] | ResNet50 | 2 | 8.7 | 53.1 | 62.6 |
| ALFNet[†] [21] | ResNet50 | 1 | 8.1 | 51.0 | 59.1 |
| RPN+BF[†] [49] | VGG16 | 2 | 7.3 | 54.6 | 59.9 |
| RepLoss[†] [14] | ResNet50 | 2 | 5.0 | 47.9 | 59.0 |
| CSP [22] | ResNet50 | 1 | 4.5 | 45.8 | **56.9** |
| KGSNet [18] | ResNet50 | 2 | 3.9 | **34.2** | **42.2** |
| JointDet [50] | ResNet50 | 2 | 3.0 | - | - |
| PedHunter [10] | ResNet50 | 2 | **2.3** | - | - |
| CSP+Proposed module | ResNet50 | 1 | **3.7** | 42.8 | 56.97 |
| | | | | | |
| ALFNet[†] [21] | ResNet50 | 1 | 4.5 | 43.4 | 56.8 |
| RepLoss[†] [14] | ResNet50 | 2 | 4.0 | 41.8 | 58.6 |
| CSP [22] | ResNet50 | 1 | 3.8 | 36.5 | 54.4 |
| CSP+Proposed module | ResNet50 | 1 | **2.6** | **28.0** | **53.9** |

generation network, the refined bounding boxes, the key-point detector and the super-resolution network. With these components, the inference speed of KGSNet is 5.9 FPS and 3.2 FPS (Titan X GPU, not including the time of using ALFNet to generate the candidate proposals) on Caltech and Citypersons datasets [18] while ours achieves 29.5 FPS and 6.8 FPS (RTX 3090 GPU) respectively. Therefore, the enhanced single-stage pedestrian detector is cost-effective with fast inference and competitive accuracy compared to complicated two-stage methods.

Table 11: Comparison with the state-of-the-art pedestrian detectors on Citypersons validation set. [†] Results reported by [22].[1] Less than 65% visibility instead of 20-65% visibility. The best and second-best results are in bold and underlined with red for two-stage methods and black for single-stage methods.

| Method | Backbone | Stage | Reasonable | Heavy | All |
|---|---|---|---|---|---|
| FRCNN[†] [45] | VGG16 | 2 | 15.4 | - | - |
| FRCNN+Seg[†][45] | VGG16 | 2 | 14.8 | - | - |
| TLL+MRF [23] | ResNet50 | 1 | 14.4 | 52.0 | - |
| OR-CNN [15] | VGG16 | 2 | 12.8 | 55.7[1] | - |
| RepLoss [14] | ResNet50 | 2 | 13.2 | 56.9[1] | - |
| ALFNet [21] | ResNet50 | 1 | 12.0 | 51.9 | - |
| CSP [22] | ResNet50 | 1 | 11.0 | 49.3 | - |
| PRNet [24] | ResNet50 | 1 | **10.8** | 42.0 | - |
| MGAN+ [17] | VGG16 | 2 | **11.0** | **39.7** | - |
| KGSNet [18] | ResNet50 | 2 | **11.0** | **39.7** | **36.2** |
| CSP+Proposed module | ResNet50 | 1 | 10.9 | **38.6** | 37.2 |

# 5 Conclusion

The paper proposes a module to enhance spatial and multi-scale features based on a single encoder of the deformable vision transformer to improve the detection accuracy of single-stage pedestrian detectors with fast inference. This module is effective on commonly used single-stage structures, including (progressively refined) anchor-based and anchor-free cases. With the CSP detection head, more than 40% parameters are reduced in the detection neck compared to CSP; however, this combination still achieves the best results in Reasonable and Heavy subsets among presented single-stage detectors on both Caltech and Citypersons datasets. Utilising pre-training, the miss rates for these subsets can be decreased to 2.6% and 28.0%, respectively, which are far better than other single-stage methods and comparable to two-stage methods. The proposed method outperforms SOTA two-stage detectors in the Heavy subset by 1.1% on Citypersons with slightly decreased inference speed. This demonstrates that single-stage detectors can be improved if spatially adaptive and multi-scale features are jointly adopted, making them cost-effective and promising. It should be mentioned that false positives appear if the attention points of a negative reference point extend to the target areas. Although the performance has been improved with the current module, the generation of attention weights and sampling locations can be carefully designed to suppress the false positives for further improvements.

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
