# A    Appendix

**Memory Cost of Self-attention Weights in DETR**: DETR has six encoder-decoder pairs. Figure 1 presents the structure of the encoder, decoder, and embedded Multi-Head Self-Attention (MHSA) layer. Each MHSA layer has a self-attention weight tensor produced by the multiplication of Query and Key as shown in Figure 1. The memory cost of this tensor during training under different hyper-parameter settings and optimization strategies are plotted in Figure 2. It shows that more attention heads, especially large downsampling ratios, significantly increase the memory cost. Additionally, Adam and AdamW optimizers, commonly used to train vision transformers, take more memory than simple SGD. For pedestrian detection tasks, we normally choose head=8, downsampling ratio=0.25 and Adam optimizer. It takes more than $10^3$GB memory to store a single attention weight tensor even when the batch size is 1. Thus, deformable DETR is used in our work to save memory resources.

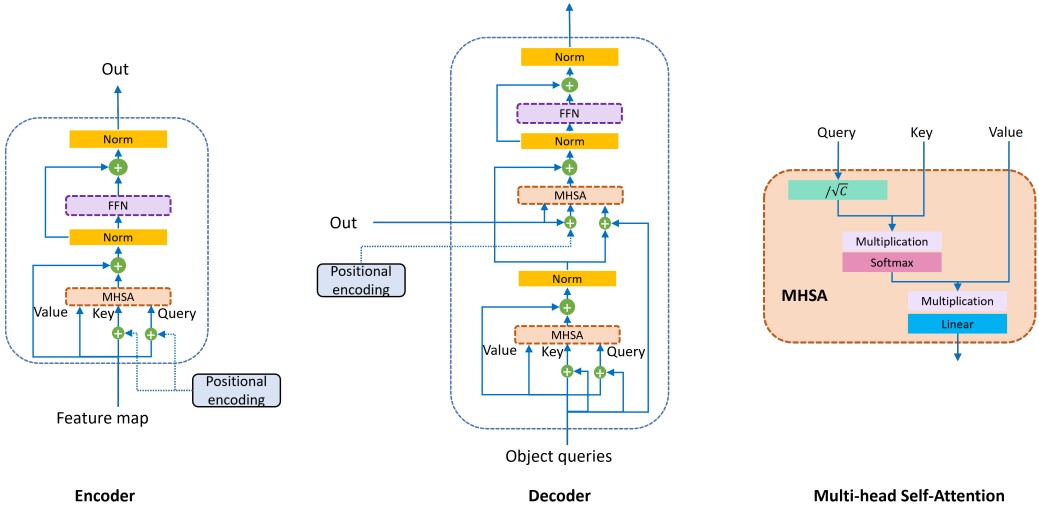

Figure 1: Architectures of the encoder, decoder, and the Multi-Head Self-Attention layer (MHSA) of DETR.

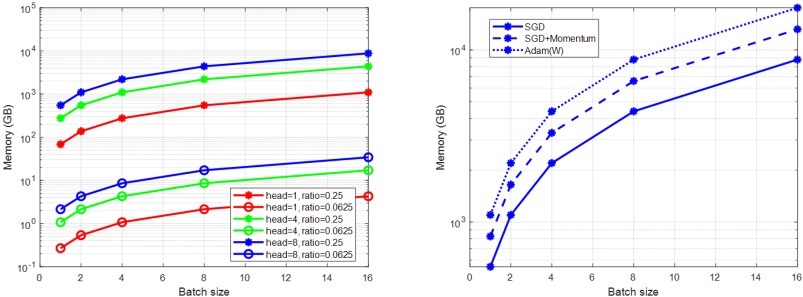

Figure 2: The memory cost of one attention weight tensor in a single multi-head self-attention layer during training. **Left**: memory cost calculated with SGD optimizer under different numbers of attention heads and downsampling ratios. **Right**: memory cost calculated with head=8, downsampling ratio=0.0625 and three different optimizers. Note that the original image size is 1024x2048. Tensors are stored as Float16 in this computation.

**Architecture**: The overall architecture of the proposed enhancement module combined with the CSP detection head is illustrated in Figure 3.

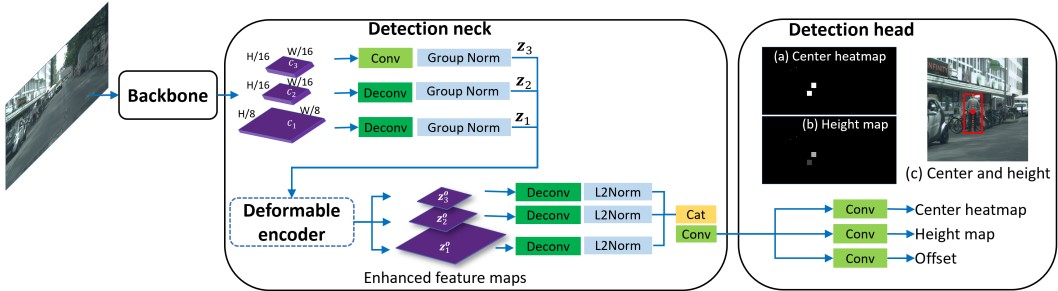

Figure 3: The overall architecture of the proposed module combined with the CSP detection head. The input images are fed into the backbone, followed by the detection neck with a deformable encoder to enhance spatial and multi-scale information. The detection head is the same as CSP. **(a)**: The ground truth center heatmap highlights pedestrian centers with white squares. **(b)**: The ground truth height map contains the logarithm of heights. **(c)**: Sketch of the center (red node) and height (orange dotted line) ground truths and their relation to the bounding box (red box) depicted on part of the input image.

**Training**: For anchor-free methods, the same ground truth and loss functions as CSP are utilized. The overall loss consists of three parts:

$$L = \lambda_c L_c + \lambda_s L_h + \lambda_o L_o \tag{1}$$

where the center heatmap loss $L_c$, height map loss $L_h$ and offset loss $L_o$ are formulated as

$$L_{center} = -\frac{1}{K} \sum_{i=1}^{\frac{W}{4}} \sum_{j=1}^{\frac{H}{4}} \alpha_{ij} \left(1 - \hat{p}_{ij}\right)^{\gamma} \log\left(\hat{p}_{ij}\right) \tag{2}$$

$$L_{height} = \frac{1}{K} \sum_{k=1}^{K} SmoothL1\left(s_k, t_k\right) \tag{3}$$

$$L_{offset} = \frac{1}{K} \sum_{k=1}^{K} SmoothL1\left(o_k, ot_k\right) \tag{4}$$

$$\hat{p}_{ij} = \begin{cases} p_{ij}, & y_{ij} = 1 \\ 1 - p_{ij}, & otherwise \end{cases} \tag{5}$$

$$\alpha_{ij} = \begin{cases} 1, & y_{ij} = 1 \\ \left(1 - M_{ij}\right)^{\beta}, & otherwise \end{cases} \tag{6}$$

where $p_{ij} \in (0, 1)$ denotes the predicted score in the center heatmap; $s_k$ and $o_k$ are the predicted height and offset at $k$-th pedestrian with the ground truth $t_k$ and $ot_k$; $M$ is a Gaussian mask map with variances proportional to the height and width of each one of the $K$ pedestrians. The two hyper-parameters $\beta$ and $\gamma$ are set as 4 and 2 empirically in accord with CSP settings. Weights $\lambda_c$, $\lambda_s$ and $\lambda_o$ for losses in equation Eq. (1) are set as 0.01, 1 and 0.1 according to CSP.

For anchor-based methods, the overall loss consists of two parts:

$$L = \lambda_{cls} L_{cls} + L_{loc} \tag{7}$$

$$L_{cls} = -\alpha \sum_{i \in S_+} \left(1 - p_i\right)^{\gamma} \log\left(p_i\right) - \left(1 - \alpha\right) \sum_{i \in S_-} p_i^{\gamma} \log\left(1 - p_i\right) \tag{8}$$

where focusing parameters $\alpha$ and $\gamma$ are set as 0.25 and 2 [1]. $S_+$ and $S_-$ stand for positive and negative sets. Smooth L1 loss is used for localization loss $L_{loc}$.

**Visualization**: To better understand how the Multi-Scale Deformable Attention (MSDA) layer works, the sampling locations and their attention weights are presented in Figure 4. The center of a single pedestrian corresponds to reference points in feature maps at the first and third scale with size

$(H/4, W/4)$ and $(H/16, W/16)$. Each reference point corresponds to learned sampling locations across eight heads and three scales with learned attention weights, which assign large values to important features. The MSDA layer accumulates the features at these locations with the weights to fuse the multi-scale and spatial adaptive information. According to Figure 4, the distribution patterns of sampling locations are similar in the presented samples. However, the differences lie in the attention weights. It is observed that high scales tend to focus on a sparser area with mild attention weights, while low scales concentrate on the area close to the reference point with relatively sharp weights. As the heat maps at a larger scale show, important sampling locations of the relatively large pedestrian seem more reasonable as the head, shoulder and leg are highlighted.

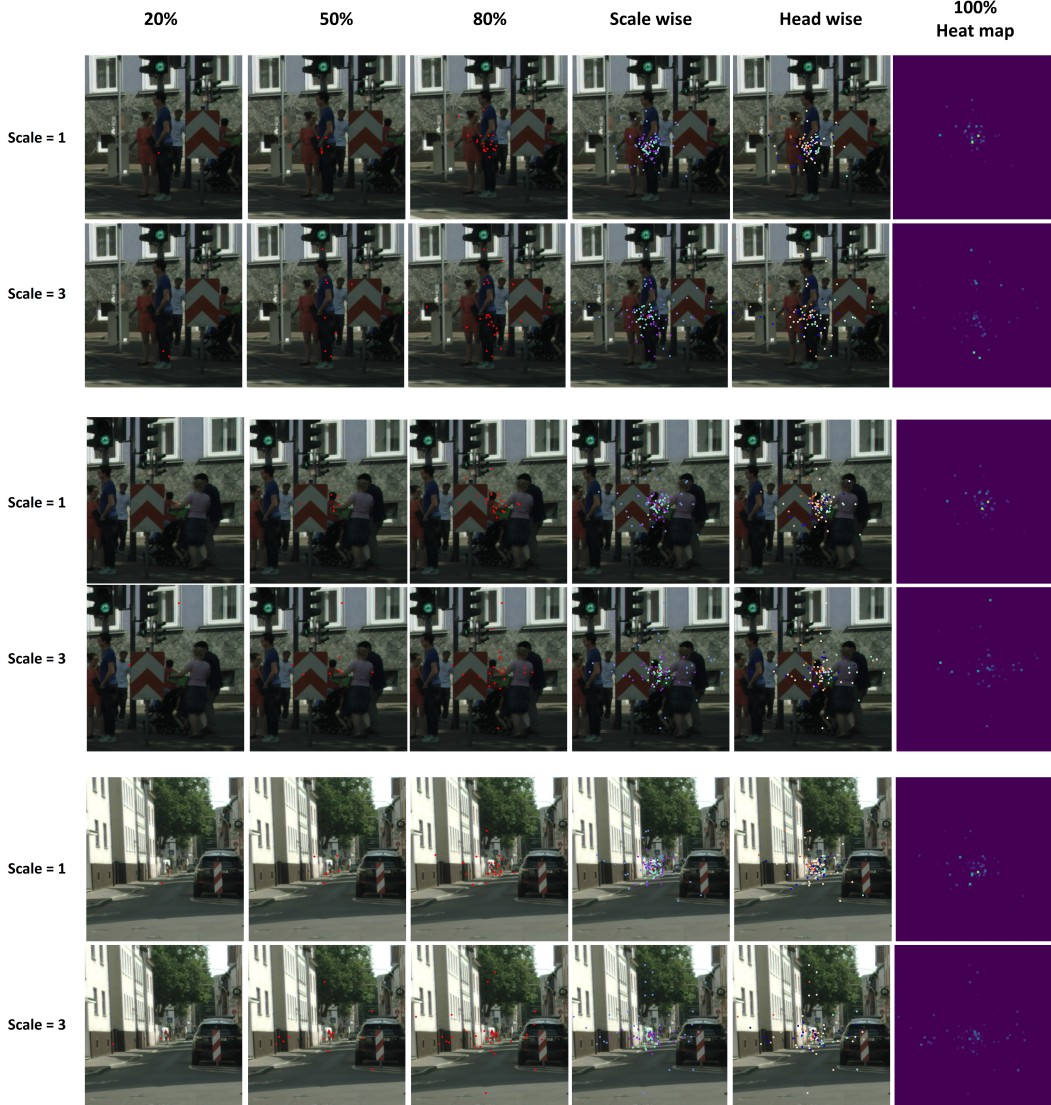

Figure 4: The visualization of corresponding attention weights and sampling locations of the pedestrian centers obtained at first and third-scale feature maps. From up to down are relatively large, significantly occluded and relatively small pedestrians. The first three columns show the locations of attention weights that account for 20%, 50% and 80%, the sum of all weights. The last column shows the heatmap of all the attention weights, the brighter the pixel, the larger the weight. The 4th and 5th columns present sampling locations of each head/scale with different colors.

Figure 5 presents some detected pedestrian samples under various circumstances, including relatively large, small, and significantly inter- and intra- class occluded ones.

**Relatively large**                    **Relatively small**

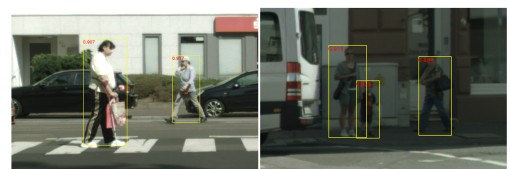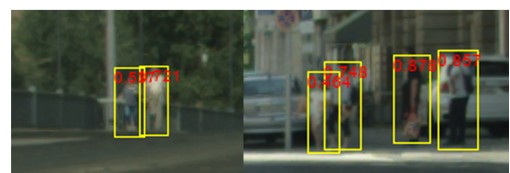

**Occluded**

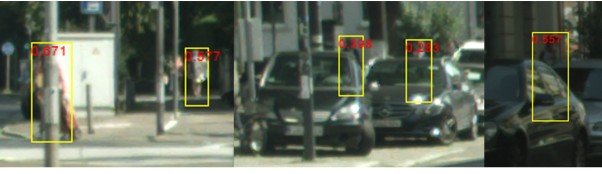

**Overlapped pedestrians**

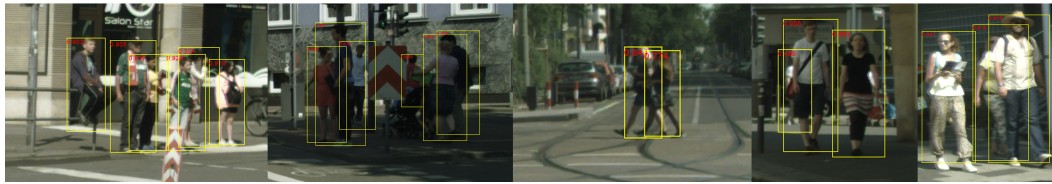

Figure 5: Different types of pedestrians from the Citypersons validation set detected by the proposed detector.

# References

[1] Tsung-Yi Lin, Priya Goyal, Ross Girshick, Kaiming He, and Piotr Dollár. Focal loss for dense object detection. In *Proceedings of the IEEE international conference on computer vision*, pages 2980–2988, 2017.