# OpenReview forum: "Effectiveness of Vision Transformer for Fast and Accurate Single-Stage Pedestrian Detection"
_NeurIPS.cc/2022/Conference — NeurIPS 2022 Accept_

### Official Review · Reviewer_w34n · 2022-07-10

**Rating:** 4
**Confidence:** 4
**Soundness:** 3 good
**Presentation:** 2 fair
**Contribution:** 2 fair

**Summary:**

The paper presents a deformable vision transformer for pedestrian detection. The vision transformer introduces multi-scale deformable attention [33] to enhance features for pedestrian classification and localization. The experiments on Caltech and CityPersons demonstrate that the enhanced features by the proposed deformable vision transformer help improve the pedestrian detection performance.

**Questions:**

The paper needs careful proofreading. It contains some grammatical errors. For example, Line 3 the latter categorys --> the latter categories or the latter category.


**Limitations:**

No. See weaknesses.

**Strengths And Weaknesses:**

Strenghts:
1. The introduced multi-scale deformable attention enhances features for pedestrian detection in one-stage pdestrian detector. The results in Tables 5 and 6 show that it brings performance improvements on both Caltech and CityPersons datasets.

2. Overall, the paper is easy to follow.

Weaknesses:
1. The novelty and contributions of the paper are limited. It simply adopts multi-scale deformable attention [33] to enhance features for pedestrian detection.
2. The performance of the proposed method is not impressive enough. On CityPersons, its peformance is only on par with state-of-the-art.

---

> ### Author Response · Authors · 2022-08-02
> **We would like to thank the reviewer for the insightful and valuable comments. Please find below our answers, point by point.**
>
> -----***Corrections are marked in blue in the paper.***-----
> # Strengths And Weaknesses:
> 1. In terms of the novelty:
>     + To the best of the writers’ knowledge, the proposed simple strategy of incorporating the vision transformers on top of the backbone of pedestrian detection tasks is a breakthrough in pedestrian detection;
>     + The trade-off between accuracy and inference speed of the single-stage pedestrian detector is achieved with only a simple but effective strategy;
>     + The idea to improve the features in spatial dimension rather than traditional channel dimension provides a new direction for the improvement of single-stage detectors.
>
>     In terms of the contributions:
>     + The main contribution is to show the effectiveness of the deformable vision transformer to achieve a trade-off between high accuracy and faster inference, which is a significant issue for single-stage pedestrian detectors. The proposed strategy is simple, but effective with the consideration of practical scenes.
>     + The idea of enhancing spatial features rather than sticking to channel features provides a new direction to enhance single-stage detectors.
>     + Based on proposed structure, even single-stage methods can make use of the techniques that used to be available only for two-stage methods, such as mask guided attentions. To this end, the proposed work is inspiring and may allow more attempts on single-stage methods.
>
> 2. Table 5, 6 show that the enhanced features do improve results on reasonable, heavy and all subsets on both Citypersons and Caltech dataset, especially in the presence of occlusion.
>     - Compared to single-stage state-of-the-arts:
>         - For Caltech dataset, the average miss rates are decreased to 3.7% and 42.8%, which are 0.8% and 3% lower than the second best single-stage detector (4.5% and 45.8%).
>         - For Citypersons dataset, the average miss rates are decreased to 10.6% and 36.7%, which are 0.2% and 5.3% lower than the second best single-stage detector (10.8% and 42.0%).
>
>         Therefore, the proposed single-stage method achieves leading performance among the single-stage state-of-the-art detectors.
>     - Compared to two-stage state-of-the-arts:
>         - For challenging Citypersons dataset, the proposed single-stage method achieves better results on reasonable and heavy subsets. It even achieves the lowest average miss rate on heavy subset (36.7%) which surpass 3% by the leading two-stage KGSNet and MGAN+(39.7%).
>         - For Caltech dataset, the proposed method further narrows the gap between single- and two- stage methods, the average miss rate for heavy subset is further decreased to 28% surpassing the second-best two-stage method by 6.2% with pretraining on Citypersons dataset. The result for reasonable subset is also decreased to 2.6% which is the second best among both single- and two- stage detectors.
>
>     These results show that the proposed single-stage detector achieves overall leading detection accuracy among all the presented state-of-the-art detectors (including second-stage methods) on Citypersons datasets and comparable accuracy on Caltech dataset.
>     It should be admitted the two-stage methods usually produce overall better accuracy than single-stage methods with the advantage of region proposal network and refined results and at the expense inference time. Apart from accuracy, fast inference also matters for pedestrian detection in practical scenes. The proposed detector has a simple structure, which is easy to perform but effective. Compared to our detector, the leading two-stage KGSNet, for example, take the advantage of additional components like proposal generation network, the refined bounding boxes, the key-point detector and the super resolution network. With these components, the inference speed of KGSNet is 5.9 frames per second (FPS) and 3.2FPS (Titan X GPU, not including the time of using ALFNet to generate the candidate proposals) on Caltech and Citypersons datasets while the proposed detector achieves 32.1FPS and 7.3FPS (RTX 3090 GPU). Considering both detection accuracy and speed, the proposed detector shows leading performance.
> # Questions:
> 1. The paper is proofread carefully.

---

### Official Review · Reviewer_6hn5 · 2022-07-11

**Rating:** 4
**Confidence:** 3
**Soundness:** 2 fair
**Presentation:** 2 fair
**Contribution:** 2 fair

**Summary:**

Due to the lack of spatially adaptive features, in this paper, the authors propose deformable vision transformer aiming to achieve the balance between speed and accuracy.

**Questions:**

1.	In page 4, line 153, what is the original work?
2.	In fig2’s caption, why don’t we need the key tensor?
3.	Can you explain the difference to [1]?
[A] Xia, Zhuofan, et al. "Vision transformer with deformable attention." Proceedings of the IEEE/CVF Conference on Computer Vision and Pattern Recognition. 2022.

**Limitations:**

They should address their limitation in conclusion or discussion section.


**Strengths And Weaknesses:**

1.	R an HO are not famous terminologies, but they explain these terms in Section 4.1. It hurts the readability.
2.	The parameters in loss function are not well-justified by experiments. The same issue is also for inference stage.
3.	Lack of simulations results in popular detection benchmarks.
4.	We already have deformable vision transformer in CVPR2022 [A].
[A] Xia, Zhuofan, et al. "Vision transformer with deformable attention." Proceedings of the IEEE/CVF Conference on Computer Vision and Pattern Recognition. 2022.
5. The performance is not very impressive.

---

> ### Author Response · Authors · 2022-08-02
> **We would like to thank the reviewer for the insightful and valuable comments. Please find below our answers, point by point.**
>
> -----***Corrections are marked in blue in the paper.***-----
> # Strengths And Weaknesses:
> 1. R and HO stand for reasonable and heavily occluded pedestrians respectively. They are commonly used experimental settings in pedestrian detection such as Pedestron [1], CSP [2]. To achieve fair comparison with other pedestrian detectors, we follow the same evaluation settings and procedures as the above-mentioned works. As for readability, we use the full terminologies (Reasonable, Heavy, Small and All) instead of abbreviations (R, HO, RS and A) in this manuscript.
> 2. The parameters in loss function loss, the uniform aspect ratio 0.41 and the score threshold in the inference stage follow the choice of the baseline CSP [2] to achieve fair comparison. According to [2], the parameters are set experimentally while the uniform aspect ratio measures the general width/height of a GT because pedestrians have similar aspect ratios. The threshold is provided by the original official code. Citation of CSP is added in the training stage and a brief explanation of the constant is added in the inference stage.
> 3. The Citypersons and Caltech pedestrian datasets are the two most used challenging datasets for pedestrian detection tasks. State-of-the-art pedestrian detectors such as Pedestron [1], CSP [2] and MGAN [3] are evaluated on these datasets. It is fair and enough to test the proposed method and compare with others conveniently on the same datasets.
> 4. [A] proposed a deformable vision transformer module and explored its application to the general backbone. It uses an offset network to produce offsets from the query tensor and take the multiplication of key and value tensors as attention weights. It produces context-aware attention weights which can extract features in the region of interest. As mentioned above, [A] focuses on the design of the deformable mechanism and applying it to establish a general backbone. While our work focuses on showing the effectiveness of the deformable mechanism to overcome significant issues (i.e., lack of accuracy) in the single-stage pedestrian detectors.
> 5. Table 5, 6 show that the enhanced features do improve results on reasonable, heavy and all subsets on both Citypersons and Caltech dataset, especially in the presence of occlusion.
>     + Compared to state-of-the-art single-stage detectors, the proposed method surpasses the second-best results by 0.8% and 3% on reasonable and heavy subsets on Caltech dataset, and 0.2% and 5.3% on Citypersons dataset.
>     + Compared to state-of-the-art two-stage detectors, the proposed method even produces better results on reasonable and heavy subsets on Citypersons. It even achieves the lowest $MR^{-2}$ on heavy subset (36.7%) which surpass 3% by the leading two-stage KGSNet and MGAN+(39.7%). For Caltech dataset, the proposed method narrows the gap between single- and two- stage methods, the $MR^{-2}$ for heavy subset is further decreased to 28% surpassing the second-best two-stage method by 6.2% with pretraining on Citypersons. The $MR^{-2}$ for reasonable subset (2.6%) achieves the second best among both single- and two- stage detectors.
>
>     Demonstrated results show that the proposed single-stage detector achieves overall leading detection accuracy among all the presented state-of-the-art detectors (including second-stage methods) on Citypersons datasets and comparable accuracy on Caltech dataset.
>     Two-stage methods usually achieve better accuracy than single-stage methods at the expense of inference time. Fast inference matters for pedestrian detection considering the practical scenes. The proposed method is a single-stage detector with a simple structure, which is easy to perform but effective. Its inference speed achieves 32.1FPS and 7.3FPS (RTX 3090 GPU) on Caltech and Citypersons. However, that of the leading two-stage KGSNet, for example, is 5.9 FPS and 3.2FPS (Titan X GPU, not including the time of using ALFNet to generate the candidate proposals). Considering both detection accuracy and speed, the proposed detector shows leading performance.
>
> [1]. Hasan, Irtiza, et al. "Generalizable pedestrian detection: The elephant in the room." CVPR. 2021.
> [2]. Liu, Wei, et al. "High-level semantic feature detection: A new perspective for pedestrian detection." CVPR. 2019.
> [3]. Pang, Yanwei, et al. "Mask-guided attention network for occluded pedestrian detection." ICCV. 2019.
>
> # Questions:
> 1. Please find citation [21]. The citation is added in Line 183, page 5.
> 2. The middle image of Figure 2 should be the multi-scale deformable attention (MSDA) layer. We correct this in the paper. In MSDA, the attention weight is obtained by embedding of the query tensor, rather than the multiplication of key and query tensors. Therefore, key tensor is not needed.
> 3. Please find point 4 in Strengths And Weaknesses.
>
> # Limitations:
> 1. The limitations have been addressed in Line 288-292, page 9.

---

> > ### Comment · Reviewer_6hn5 · 2022-08-08
> > **The novelty and contributions of the paper are limited**
> >
> > I agree with Reviewer w34n  that the novelty and contributions of the paper are limited. The key contribution of this paper is applying Deformable Vision Transformer [A] in SSD and the idea of multi-scale deformable attention is also similar to [33].  I tend to remain the same rating.

---

> > > ### Author Response · Authors · 2022-08-08
> > > **We would like to thank the reviewer for the feedback. Please find the following response.**
> > >
> > > - The key contribution is showing the effectiveness of the deformable vision transformer to address a significant issue in single-stage detectors which is the lack of accuracy.
> > > - Our motivation is to address the lack of spatially adaptive features in single-stage pedestrian detectors. It is proved that this can be done with a simple use of only one encoder of the vision transformer, and no other complicated structures are needed. This simpleness serves as a positive point of the paper as it is easy to perform and effective.
> > > - There are very few works applying vision transformer to the pedestrian detection task. [37] has proved that using DETR directly in pedestrian detection produces worse result than the baseline Faster R-CNN, this suggests that how to apply the vision transformer on top of the backbone remains a problem. To this end, this paper carefully designed the use of the vision transformer (Table 3, 4), rather than applying [35] directly to achieve the best results with least inference time and minimum parameters. To the best of the writers’ knowledge, the paper presents the first solution of using vision transformers on top of the backbone for the pedestrian detection society.
> > > - The paper is based on [35] (not [33], citations are in current version) which is published earlier than [A]. Both [35] and [A] proposed the deformable vision transformer, but [35] predicts attention weights and sampling offsets with the query tensor via a simple linear embedding, [A] remains the same attention weight generation method as traditional transformers in NLP and uses a network to predict offsets. Compared to [A], the attention mechanism in [35] requires less parameters and is more efficient which is suitable to our need.
> > >
> > > We briefly summarize the novelty and contributions as follows:
> > > In terms of the novelty:
> > > + To the best of the writers’ knowledge, the proposed simple strategy of incorporating the vision transformers on top of the backbone of pedestrian detection tasks is a breakthrough in pedestrian detection.
> > > + The trade-off between accuracy and inference speed of the single-stage pedestrian detector is achieved with only a simple but effective strategy.
> > > + The idea to improve the features in spatial dimension rather than traditional channel dimension provides a new direction for the improvement of single-stage detectors.
> > >
> > > In terms of the contributions:
> > > + The main contribution is to show the effectiveness of the deformable vision transformer to achieve a trade-off between high accuracy and faster inference, which is a significant issue for single-stage pedestrian detectors. The proposed strategy is simple, but effective with the consideration of practical scenes.
> > > + The idea of enhancing spatial features rather than sticking to channel features provides a new direction to enhance single-stage detectors.
> > > + Based on proposed structure, even single-stage methods can make use of the techniques that used to be available only for two-stage methods, such as mask guided attentions. To this end, the proposed work is inspiring and may allow more attempts on single-stage methods.

---

### Official Review · Reviewer_pH8n · 2022-07-11

**Rating:** 6
**Confidence:** 2
**Soundness:** 2 fair
**Presentation:** 3 good
**Contribution:** 3 good

**Summary:**

Pedestrian detection is crucial in many applications, but the fluctuation of information from the images makes it difficult to do it accurately. CNN-based two-stage detectors are used for this task but are slow at inference and are difficult to train. Single-stage detectors solve these issues but have a higher miss rate, especially in the presence of occlusion. The authors suggest that these problems stem from information fed to the detection head: single-stage detectors get insufficient information compared to two-stage detectors.

To solve the above-mentioned problems, the authors proposed to use a deformable vision transformer. The transformer enhances multi-scale features extracted from a backbone, thereby providing better information to the detection head. They also proposed to use center-and-scale prediction (CSP). The proposed method uses multi-scale features extracted from a CNN. These features are passed through a detection neck. These features are then passed through a deformable transformer to enhance them. These enhanced features are then used to find three outputs that are relevant to CSP. Comprehensive experiments are performed to show the performance of the method on two pedestrian datasets. A set of ablation studies are also conducted to show the effect of different components of the method.

**Questions:**

## Questions / Weaknesses

1. What is the main contribution of the paper. More specifically, what parts of the method are proposed. Or the main contribution is to show the effectiveness of vision transformers in overcoming the issues in single-stage detectors? Both contributions (proposing a new method or showing the effectiveness of existing ones) are equally important, but this should be highlighted clearly.

2. If I understand it correctly, the main proposed components are the use of deformable transformers to enhance features and the use of CSP.

3. If I'm not mistaken, similar architectures are also used in general object detection. It would be interesting to highlight the difference between general object detection and how the proposed method is different.

4. In the relevant work, it is noted that DETRs don't perform better than two-stage-detectors. Is this because DETR produces a certain number of encoded outputs? It would be great if the authors can comment on it.

**Limitations:**

The limitations of the work are not highlighted clearly.  It would be interesting to see some cases where the proposed method fails.

**Strengths And Weaknesses:**

## Strengths

1. The paper is mostly well written and easy to follow.

2. The motivation, to bring the speed of inference in single-stage detectors by using features enhanced by transformers, is interesting.

3. The well-motivated problem is then solved in a way that makes sense.

4. Results (in Table 5, 6) show that enhanced feature maps do improve results, especially in the presence of occlusion.

5. Comprehensive experiments are performed to show not only the SOTA results but also the effect of individual components.

Weaknesses are pointed out in the next section.

---

> ### Author Response · Authors · 2022-08-02
> **We would like to thank the reviewer for the insightful and valuable comments. Please find below our answers, point by point.**
>
> -----***Corrections are marked in blue in the paper.***-----
> # Questions:
>
> 1. Thank you for this comment. The main contribution is to show the effectiveness of the deformable vision transformer to improve single-stage detectors. We include this in the main contributions in the Introduction in Line 89-90, page 2.
> 2. Yes.
> 3. The differences are twofold.
>     + The pedestrians are often heavily occluded, with significantly varying scales and appear as crowds as presented in pedestrian datasets while the targets are less challenging in general object detection datasets, such as COCO. To overcome these challenges, methods for pedestrian detection usually make use of the unique characteristics of pedestrians, such as key point and visible parts which may not be suitable for general objects. In the proposed method, this would be the aspect ratio, but fortunately it does not influence the architecture. Therefore, the proposed network can be used to general object detection by adding a parallel branch to predict the width of the bounding box. It can also be used for classification with a classification head.
>    + Pedestrian detection datasets are usually smaller and not as informative as general object detection datasets as they only have a single category. The size of the proposed network may be smaller than that of the general methods such as DETR. It might be possible that the proposed network is not complicated enough to learn the mapping of multiple objects and corresponding GTs.
>
> 4. We assume ‘DETRs’ refers to DETR and deformable DETR. Because training DETR is extremely time consuming compared to the deformable one, the relevant work focuses on the latter. So, please allow us to narrow to the deformable DETR when addressing this question. The relevant work suggests that the decoder part of the deformable DETR results in the poor performance in pedestrian detection for two reasons:
>     + The mapping from sparse uniform object queries in the decoder to local dense pedestrian clusters is ambiguous and misses GTs;
>     + The attention positions in the decoder do not cover the pedestrians well. They are too compact/extensive for a large/small target.
>
>     We think that the reasons provided by the relevant work make sense, particularly, weak attention positions (point 2) are also observed in our network (see Figure 4 in the supplementary materials) in the encoder part. We believe that focusing sampling locations and corresponding attention weights on targets properly could further improve the vision transformer-based pedestrian detection; similar to Mask Guided Attention Network (MGAN) which improves Faster R-CNN by enhancing the features at visible parts while suppressing the features at the background.
>     The number of encoded outputs for each encoder is the sum of pixels ($H\times W$) of feature maps at multiple scales. As it retains the same through multiple encoders, the resolution of the enhanced feature map is retained which reserves the details for detection of relatively small targets. With more scales and proper resolution (such as including both higher- ($H/16\times W/16$) and lower-level ($H/4\times W/4$) feature maps) at each scale, the performance of the deformable DETR can be improved. But not sure if this could allow it to surpass two-stage detectors like Faster R-CNN.
>     Apart from the above reasons, overfitting is observed in training the deformable DETR on 12.5%, 25% and 50% of the Citypersons dataset. The losses on the validation set begin to increase after approximately 7, 10 and 15 epochs. This implies that the network might be too complicated for pedestrian datasets. It might be better to train with simpler structures or more prior knowledge.
>
> # Limitations:
> 1. As mentioned in Question 4, one limitation is that currently the sampling locations and attention weights do not cover the target evenly especially when it is too large or small, which may miss GTs. Another significant problem is that false positives are introduced if the attention points of a negative reference point extend to the target areas. The generation of attention weights and sampling locations can be carefully designed to achieve further improvements. We have added the limitations in Line 288-292, page 9.

---

### Official Review · Reviewer_s2Bv · 2022-07-12

**Rating:** 6
**Confidence:** 4
**Soundness:** 3 good
**Presentation:** 3 good
**Contribution:** 3 good

**Summary:**

The paper addresses the crucial problem in computer vision: pedestrian detection. The issue of faster inference (i..e real-time performance) with competitive accuracy is still a major problem as stated by the author(s). To solve this major issue/problem, the author(s) proposes a single-stage anchor-free pedestrian detector with the deformable vision transformer to balance the issue of faster inference and high accuracy. Together with strong experiments, solid baselines, and evaluation criteria(s), the author(s) achieves competitive results on well-known datasets of Caltech and Citypersons.

**Questions:**

There is a major concern, I have on this manuscript's contribution that although the trade-off between higher accuracy and faster inference has been seriously taken by the author(s), there is little-to-less information has been provided on the occlusion part. Well, in general, if we narrow it down to common scenarios, we see an occluded person(s) in lot more cases and should be a concern to raise, whereas except for related work, I wasn't able to see major information on this issue. I appreciate the author's efforts in providing very clear information on their contribution, could I please ask the authors to also provide some insight on this case.

**Limitations:**

There's a significant limitation in terms of providing insight on the occlusions/occluded person(s). However, I do not see any major limitations except the mentioned question, the proposed solution for a trade-off between high accuracy and faster inference using Vision Transformer is a pretty solid contribution, in my view to the computer vision community.

**Strengths And Weaknesses:**

++ Novelty

- The task formulation is concise, convincing, solid, and novel. A seemingly reasonable approach has been conducted in this paper. Compared to the existing single/two-stage detectors, the proposed single-stage detector based on the feature extractor and CSP strategy achieves competitive results.
- To the best of my knowledge, this simple strategy of incorporating the vision transformers on top of the backbone of pedestrian detection tasks is a solid contribution to the computer vision community.

++ Evaluation

- The experiments are strong, sufficient, and very well presented. The analysis presented here showcases the author's efforts on how the small things have been noted down while performing such extensive experiments with solid evaluation criteria(s).
- The experimental evaluations demonstrate the effectiveness of the complete approach and showcase its practical value. The detailed analyses clarify the contribution of each component.

++ Clarity

 - The proposed solution by providing a multi-scale deformable feature extractor based on a deformable vision transformer is theoretically sound and well-written clearly describes the improvements and adequately contextualizes the contributions.
- The manuscript also provides a good description of related work and background, motivating the problem. Additionally, it provides a thorough description of the part of the pipeline it deals with.

---

> ### Author Response · Authors · 2022-08-02
> **We would like to thank the reviewer for the insightful and valuable comments. Please find below our answers, point by point.**
>
> -----***Corrections are marked in blue in the paper.***-----
> # Questions:
> 1. Thank you for mentioning that limited information of the occlusion part is included in the main content. Due to the limited pages of the main content, the samples of detected occluded pedestrians are presented in Figure 3 in the supplementary materials. These samples cover both the intra- and inter- occlusion cases. Apart from detection results, we also visualize the sampling locations and attention weights at multiple scales of an occluded pedestrian in the mid group of Figure 4 in the supplementary materials.
> Also, to address this comment, we have added an explanation of how the proposed method improves the detection of occluded pedestrians in Line 248-252, page 7 in the main content.
>
> # Limitations:
> 1. The limitations have been addressed in Line 288-292, page 9 in the main content and in the supplementary materials.

---

### Meta-Review · Area_Chair_ur54 · 2022-08-24

**Recommendation:** Accept
**Confidence:** Certain

**Metareview:**

This paper improves single-stage pedestrian detectors with deformable vision transformers.
The paper initially received mixed reviews, *i.e.* two weak accept and two bordeline reject recommendations. The reviewers' main concerns were essentially related to the novelty and contributions of the submission and to the performance's assessment of the proposed approach. The rebuttal provided elements to clarify the scope and contributions of the paper, but R6hn5 still challenges the novelty.

The AC's owns reading of the submission leads to the following analysis:
- The AC agrees that the novelty can be challenged, in the sense that the submission uses known components, namely deformable transformers and Center and Scale Prediction (CSP), in the context of pedestrian detection. Although the paper is overall well written, a more coarse-to-fine presentation of the approach, starting from the description of the overall pipeline in Figure 3, and then describing the proposed multi-scale deformable attention (MSDA) and CSP would have help the reading flow in AC's opinion.
- On the other hand, the paper is clearly motivated and the adaptation of both components have not been successfully applied to pedestrian detection yet. The approach shows that specific architecture design, *e.g.* with a smaller number of encoders, is required to make the approach successful. The combination of MSDA and CSP in this context is also meaningful and experimentally validated in ablation studies. Finally, the absolute performances reached by the approach is convincing, by reducing the gap between single-stage approaches and two-step methods, and even outperforming state-of-the-art performances in several cases.

Therefore, the AC recommends paper acceptance. It highly encourages the authors to take into account reviewers and AC remarks to improve the final paper.

**Award:**

No

---

### Decision · Program_Chairs · 2022-09-14

Accept